# Pregnancy after Kidney Transplantation—Impact of Functional Renal Reserve, Slope of eGFR before Pregnancy, and Intensity of Immunosuppression on Kidney Function and Maternal Health

**DOI:** 10.3390/jcm12041545

**Published:** 2023-02-15

**Authors:** Rebecca Kaatz, Elisabetta Latartara, Friederike Bachmann, Nils Lachmann, Nadine Koch, Bianca Zukunft, Kaiyin Wu, Danilo Schmidt, Fabian Halleck, Peter Nickel, Kai-Uwe Eckardt, Klemens Budde, Stefan Verlohren, Mira Choi

**Affiliations:** 1Department of Nephrology and Medical Intensive Care, Charité—Universitätsmedizin Berlin, Corporate Member of Freie Universität Berlin and Humboldt-Universität zu Berlin, 13353 Berlin, Germany; 2Department of Obstetrics, Charité—Universitätsmedizin Berlin, Corporate Member of Freie Universität Berlin and Humboldt-Universität zu Berlin, 13353 Berlin, Germany; 3Tissue Typing Laboratory, Charité—Universitätsmedizin Berlin, Corporate Member of Freie Universität Berlin and Humboldt-Universität zu Berlin,13353 Berlin, Germany; 4Department of Pathology, Charité—Universitätsmedizin Berlin, Corporate Member of Freie Universität Berlin and Humboldt-Universität zu Berlin, 13353 Berlin, Germany

**Keywords:** kidney transplantation, pregnancy, renal reserve capacity, eGFR slope

## Abstract

Women of childbearing age show increased fertility after kidney transplantation. Of concern, preeclampsia, preterm delivery, and allograft dysfunction contribute to maternal and perinatal morbidity and mortality. We performed a retrospective single-center study, including 40 women with post-transplant pregnancies after single or combined pancreas–kidney transplantation between 2003 and 2019. Outcomes of kidney function up to 24 months after the end of pregnancy were compared with a matched-pair cohort of 40 transplanted patients without pregnancies. With a maternal survival rate of 100%, 39 out of 46 pregnancies ended up with a live-born baby. The eGFR slopes to the end of 24 months follow-up showed mean eGFR declines in both groups (−5.4 ± 14.3 mL/min in pregnant versus −7.6 ± 14.1 mL/min in controls). We identified 18 women with adverse pregnancy events, defined as preeclampsia with severe end-organ dysfunction. An impaired hyperfiltration during pregnancy was a significant risk contributor for both adverse pregnancy events (*p* < 0.05) and deterioration of kidney function (*p* < 0.01). In addition, a declining renal allograft function in the year before pregnancy was a negative predictor of worsening allograft function after 24 months of follow-up. No increased frequency of de novo donor-specific antibodies after delivery could be detected. Overall, pregnancies in women after kidney transplantation showed good allograft and maternal outcomes.

## 1. Introduction

Fertility after successful kidney transplantation (KTX) improves dramatically compared to women with chronic kidney failure [1,2]. Live birth rates in post-transplant pregnancies range from 72 to 80% as demonstrated by studies from registries and retrospective cohorts [3,4,5].

Of concern, preeclampsia, preterm delivery, and infants born with small gestational age contribute to maternal and perinatal morbidity and mortality [6,7]. Amongst others, the known risk factors for unfavorable pregnancy outcomes are advanced impaired kidney function, uncontrolled hypertension, pre-existing proteinuria, a history of ≥2 kidney transplants, and immunosuppressive drugs at the time of conception [8,9,10]. Pregnancy itself may lead to a decline in kidney function, an increase in proteinuria, and hypertension.

Patients after KTX have underlying and ongoing chronic kidney disease (CKD), and it can be difficult to discriminate whether the cause of the deterioration in allograft function is due to pregnancy or due to further progress of CKD. Moreover, in women after KTX, the percentage of patients with preeclampsia might have been misinterpreted using the previous classic criteria for preeclampsia according to the ISSHP (international society for the study of hypertension in pregnancy) guidelines, such as hypertension greater than 140/90 mm Hg and proteinuria >300 mg/day after 20 weeks of gestation [11]. As a consequence, the risk factors contributing to preeclampsia in pregnant KTX patients remain difficult to define. Placental dysfunction as seen in preeclamptic pregnancy is associated with an imbalance in angiogenic and antiangiogenic factors, including placental growth factor (PlGF) and soluble fms-like tyrosine kinase-1 (sFlt-1) [12,13,14]. Thus, the measurement of angiogenic markers, either alone or combined as part of the sFlt-1/PlGF ratio, has significant value in preeclampsia prediction [15,16,17] but is poorly investigated in women with post-transplant pregnancy. Likewise, screening for altered uteroplacental and fetoplacental perfusion using ultrasound doppler is routinely performed, but abnormal ultrasonographical findings in transplanted patients are poorly defined.

Here, we retrospectively analyzed risk factors contributing to adverse pregnancy outcomes in women after kidney transplantation. We took into account recent changes regarding the definition of preeclampsia as new onset of hypertension and proteinuria or new onset of hypertension and/or significant end-organ dysfunction with or without proteinuria after 20 weeks of gestation [18]. In addition, we compared the outcomes of kidney function with a matched-pair cohort of transplanted patients without pregnancy from our center at the Charité to address the question if changes in allograft function depict the consequence of pregnancy itself rather than non-pregnancy-related risk factors.

## 2. Materials and Methods

### 2.1. Patient Selection and Data Acquisition

In this retrospective single-center study, we included women with post-transplant pregnancies that started between 2003 and 2019 and who were regularly followed up in our transplant center at the Charité. Patients had received a kidney donation after brain death or a living donor kidney transplantation, except three of the women with a combined pancreas–kidney transplantation. Pregnancies were confirmed with a positive screening test and/or a positive serum ßHCG. All pregnancies resulting in birth, stillbirth, or miscarriages were included. From 63 identified pregnancies, 17 were excluded due to missing data or a lack of sufficient follow-up. Of note, data from 17 kidney transplant recipients (KTR) were published previously by Bachmann et al. [19]. However, detailed risk factors for adverse pregnancy and allograft outcomes and correlation to fetal sonographic findings were not analyzed previously, which is why we included data from these patients in the present study. Immunosuppression was modified in the case of a planned pregnancy: mycophenolic acid (MPA) was replaced with steroids or azathioprine, respectively. In the case of an unplanned pregnancy, the immunosuppressive regimen was modified immediately after confirmation of pregnancy.

We selected the following primary endpoints regarding maternal, fetal, and allograft outcome: first, the occurrence of an adverse pregnancy event (APE), defined as severe features of preeclampsia such as severe hypertension (>160/110 mmHg) and/or specific signs or symptoms of significant end-organ dysfunction, namely acute kidney injury (AKIN) level II or III, changes in laboratory parameters such as thrombocytopenia, hemolytic anemia, or organ dysfunction without alternative explainable reasons; and second, abortion ≥ 12 weeks, stillbirth, early preterm delivery ≤ 32 weeks of gestation, intrauterine growth restriction (IUGR), defined as early placental insufficiency with intrauterine growth retardation (growth <10th percentile and pathologic doppler-ultrasound of umbilical or uterine arteries). Second, we selected the allograft’s outcome as another primary endpoint, namely the occurrence of kidney failure defined as renal graft loss, or deterioration of estimated glomerular filtration rate (eGFR) ≥ 5 mL/min at 24 months follow-up after the end of pregnancy compared to the mean pre-pregnancy eGFR. If available, sFlt-1 and PlGF were assessed during the second and third trimesters of pregnancy and analyzed as the sFlt-1/PlGF-ratio. In the case of repeated measurements, the highest value of each trimester was taken into account.

The following factors were analyzed for risk associated with adverse maternal, fetal, or renal outcomes: time from transplantation, patient age, donor age, type of donation, systolic and diastolic blood pressure, pre-existing diabetes mellitus, immunosuppressive regimen, donor-specific antibody (DSA) positivity, amount of proteinuria, eGFR slope up to 24 months before, at the time of the start of and during pregnancy. Proteinuria was expressed as mg/g creatinine when available (otherwise set equal to mg per day for previous measurements). The Chronic Kidney Disease Epidemiology Collaboration (CKD-EPI) formula was used to calculate the eGFR rates. The time interval from transplantation to pregnancy was set to the estimated date of conception, and in the case of several pregnancies, it was calculated separately for each pregnancy. Gestational age was calculated in weeks starting from the first day of the last menstrual period. In the presence of the first-trimester ultrasound, gestational age was calculated according to crown length. In the case of routine screening appointments at our center, fetal sonography was performed at least once every trimester of pregnancy. An experienced physician investigated patients with standardized ultrasound procedures. Mean pre-pregnancy eGFR was determined using the mean of at least two eGFR measures prior to conception (around −6, −12 months, and around the first day of the last menstrual period). The transplant outcome was followed until 31 December 2021. For data collection, our web-based electronic patient database “TBase” [20] and clinical charts yielding complete data sets were used. Neonatal outcome was assessed including birth weight, height, and gestational age.

### 2.2. Matched-Pair Analysis

In order to generate a matched control group, KTX patients without pregnancies during the observation period were identified using TBase in analogy to the pregnancy cohort. Patients were 1:1 matched for sex, age at transplantation (±3 years), type of donation, time from transplantation to pregnancy (±1 year), eGFR ± 5 mL/min, and proteinuria ± 100 mg/g creatinine (or 100 mg/day). For women with multiple pregnancies, only the initial post-transplant pregnancy was matched. Follow-up times started after the end of pregnancy or at an equal time span in the matched-pair control.

### 2.3. Statistics

Continuous variables were expressed as mean ± standard deviation (SD) or median and interquartile range (IQR) according to their distribution. Statistical analyses were performed using GraphPad Prism Version 9 and IBM SPSS statistics version 28.0. Measurements were tested for normal/lognormal distribution prior to analysis. Differences between groups that deviated significantly from the Gaussian distribution were assessed using the Mann–Whitney U test (two groups) and paired samples (repeat measurements) were compared using the Wilcoxon matched-pairs signed rank test. *p* values less than 0.05 were considered significant. Qualitative outcomes of different cohorts were assessed using the Chi-square test with Yates’ correction. All 95% confidence intervals for proportions were calculated using the Wilson procedure with a correction for continuity. The survival probability at follow-up was calculated using the Kaplan–Meier method. Log-rank tests were used to compare survival between the different groups.

## 3. Results

Between 2003 and 2019, 43 pregnancies after single and three pregnancies after combined pancreas–kidney transplantation were included in our analysis. The total number of women was forty, of whom six had two pregnancies during the study period. Pre-pregnancy baseline characteristics and kidney function parameters of women with post-transplant pregnancies and matched-paired controls are depicted in Table 1. Between both groups, there were no significant differences in the age at pregnancy, age at transplantation, time from kidney failure to transplantation, and the time between transplantation and the start of pregnancy (or to the start of observation in the case of controls). Furthermore, the percentage of living donation, baseline allograft kidney function, and amount of proteinuria did not differ.

Compared to women with post-transplant pregnancies (pregnancy group, PG), women in the control group (CG) had slightly, but significantly, older organ donors and a higher proportion of maintenance therapy with triple immunosuppression, most of them using corticosteroids (CS), MPA, and a calcineurin inhibitor (CNI). The causes of kidney failure before transplantation were predominantly glomerulonephritis in both groups. Regarding immunological risks, the pre-existence of donor-specific antibodies was low in both cohorts, namely two women in the PG versus four women in the CG.

In the PG, eight women had a previous post-transplant pregnancy, six experienced pregnancy while being on hemodialysis therapy, and 15 reported a previous miscarriage. Six of the women became pregnant after their second KTX and one woman after her third KTX. Most of the cases were planned pregnancies, thus a change in immunosuppressive medication in advance was performed in 34 women, whereas 10 women did not change their immunosuppressive regimen due to dual therapy with CS and CNI. Two women had unplanned pregnancies with ongoing MPA intake. Both resulted in an abortion at gestational weeks five and twelve.

### 3.1. Pregnancy Outcome

A total of 39 out of 46 pregnancies ended up with a live-born baby, and the median gestational week at delivery or at the end of pregnancy was 35.15 weeks (IQR 31.5, 37.225). The reasons for a negative outcome were two early abortions before week 12 with exclusion from further analysis, one abortion at week 12, two induced abortions at weeks 19 and 22 due to severe fetal malformations, one abortion due to early bubble rupture at week 20 and one stillbirth due to severe preeclampsia at gestational week 25. Detailed adverse pregnancy outcomes and kidney function at the end of pregnancy are listed in Table 2.

### 3.2. Maternal Outcome

We identified 18 pregnancies with adverse pregnancy (APEs) events defined as preeclampsia with severe features (as described in the Methods Section). To analyze the risk factors, we divided the PG by status for adverse events during pregnancy (Table 3). Within our cohort, frequencies of hypertension, diabetes, previous pregnancies, or miscarriages did not correlate with adverse pregnancy outcomes. Next, we analyzed if impaired kidney function was associated with a negative pregnancy outcome. This was not the case as kidney function at the start of pregnancy did not differ significantly between pregnancies with and without adverse events (eGFR of 61 ± 22 vs. 64 ± 22 mL/min, respectively, Table 1). Next, we analyzed, whether an allograft’s renal reserve capacity, measured as the amount of maximal hyperfiltration during pregnancy compared to pre-pregnancy eGFR, was more relevant than an absolute cut-off value. We also analyzed whether a higher renal reserve was beneficial for better pregnancy and kidney outcome, and vice versa, or whether a diminished renal reserve capacity during pregnancy posed a risk for an APE. We observed a significant difference regarding the allograft’s renal reserve capacity during gestation between women without APE (+16.6 ±11.3 mL/min) and women with APE (+7.8 ± 13.9 mL/min, *p* < 0.05, Figure 1A, Table 3). Along the same line, the mean percentage of eGFR increase during pregnancy in relation to pre-pregnancy eGFR was 25 ± 17% in women without APE versus 10 ± 24% in women with APE (*p* < 0.01, Figure 1B, Table 3). In addition, the overall number of allografts with a >20% increase in eGFR from pre-pregnancy eGFR differed significantly between both groups (65% in women without APE versus 17% in women with APE, *p* < 0.01). The amount of proteinuria increased markedly during pregnancy (Table 3) but did not differ between pregnancies with and without APE (*p* = 0.476, Figure 1C). We observed a weak positive correlation between better pre-pregnancy eGFR and the length of gestational weeks, and between higher renal filtration reserve of the allograft and the length of gestational weeks (Appendix A). Regarding the intensity of immunosuppressive regimens, we observed a significantly higher amount of triple immunosuppression during pregnancy in women with adverse pregnancy events, which mainly consisted of CS, CNI, and azathioprine (Table 3). Serial fetal ultrasound measurements of uterine and umbilical arteries showed higher pulsatility indices (PI) of the uterine artery during weeks 24–26 in women with APE (*p* < 0.05, Table 3).

Finally, the values of the sFlt-1/PlGF-ratio (soluble fms-like tyrosine kinase-1/placental growth factor ratio) were available from 12 pregnancies during the 2nd trimester and 14 pregnancies during the 3rd trimester, and the mean values were, as expected, higher during the 3rd trimester. Women with APE showed a trend in a higher sFlt-1/PlGF ratio during the second trimester, but due to the low numbers of available measurements and large overlap, its usefulness as a risk predictor for pre-eclampsia and/or APE could not be analyzed sufficiently (Appendix A).

### 3.3. Allograft Function

As already pointed out, kidney allograft function at the start of pregnancy did not differ between the PG and CG (Table 1). Noteworthy, the eGFR slope from mean pre-pregnancy/matched pre-observation eGFR to the 24 months follow-up (FU) period after the end of pregnancy/matched observation showed a mean eGFR decline in both groups (−5.4 ±14.3 mL/min in the PG versus −7.6 ± 14.1 mL/min in the CG, n.s., Figure 2A, Table 4). The annual decline in kidney function was −3.4% in the PG versus −4.6% in the CG. Similarly, no significant differences were obtained regarding the mean levels of proteinuria between baseline and 24 months FU, although mean proteinuria increased significantly during pregnancy and was significantly higher at the end of pregnancy, defined as the timepoint of delivery or interruption of pregnancy (e.g., abortion) (1051 ± 1541 vs. 205 ± 225 mg/creatinine in PG vs. CG, respectively, Figure 2B, Table 4, *p* = 0.011). To analyze the risk factors for the deterioration in graft function over time, independent of pregnancy, we analyzed both cohorts together, the PG and CG, stratified by status for deterioration in eGFR (eGFR decrease >5 mL/min) by 24 months after the end of pregnancy or a matched control time point (Table 5). We hypothesized that a negative eGFR slope instead of an absolute eGFR value right before pregnancy/observation might be more suitable to determine stable or worsening kidney function over time. Therefore, we analyzed eGFR slopes at 24, 18, and 12 months before the start of pregnancy/observation in all KTR and correlated this with eGFR at the 24 months FU endpoint (Figure 2C, Table 5). Only the eGFR slope at 12 months before pregnancy differed significantly between women with stable (mean eGFR slope of 4.9 ± 12.0 mL/min) and worsening (mean eGFR slope of −3.9 ± 11.0 mL/min) kidney function. Thus, a negative eGFR slope 12 months before pregnancy/observation was associated with a risk of further decline in kidney function over time, independent of the status of pregnancy (*p* < 0.01). Furthermore, more allografts with a stable eGFR course over time showed the capacity for hyperfiltration, namely, to increase eGFR during pregnancy >15 mL/min or >20% from baseline eGFR (Table 5), which was in line with the observation that a higher renal reserve capacity correlated with a positive pregnancy outcome. We did not observe relevant differences regarding the increase in proteinuria during pregnancy between stable and worsening eGFR (Figure 2D).

We analyzed if pre-pregnancy eGFR, a longer time interval since kidney transplantation, and higher donor age at transplantation influenced the allograft’s renal reserve capacity but did not observe significant correlations (Appendix A).

### 3.4. Long-Term Follow-Up and Allograft Loss

Patient survival in all KTR was 100% during the observed period. The median (IQR) range of follow-up after delivery or the end of pregnancy was 75.5 (49.5, 135.3) months with graft failures in seven women after a median time of 5.6 (3.0, 19) years. The median range of follow-up in the control cohort was 74.5 (43.5, 131.0) months with graft failures in five women after a median time of 13.0 (11, 17.5) years. The difference between graft loss in the PG and CG was not significant, as described by the Kaplan–Meier estimator shown in Figure 3 (*p* = 0.467). The main causes for allograft loss and the histopathologic results of the allograft biopsy according to the BANFF2017 classification histology scores are summarized in Appendix A. Four out of seven patients in the PG developed DSA compared to 3/5 graft losses in the control group. Interestingly, de novo DSA occurred more than 1.5 years after pregnancy in one patient and in more than seven years in the other three patients. We experienced one very early pregnancy-related graft loss two months after delivery. This woman received her kidney transplant 22 years ago, and her allograft function at the start of pregnancy was marginal (12 mL/min) due to chronic allograft nephropathy with fibrosis (Appendix A). Pre-existing proteinuria of 500 mg/g creatinine worsened up to 4 g/g creatinine. Two months after delivery, dialysis therapy was initiated due to persistent allograft failure.

Overall, we did not observe more graft loss due to rejection episodes in women with pregnancy compared to the CG.

## 4. Discussion

In this study, we analyzed outcomes in women with post-transplant pregnancies using a single center cohort regarding the mothers’ health and kidney allograft function.

The main contributors to a worse pregnancy outcome were a negative eGFR slope during the last 12 months before the start of pregnancy, maintenance of triple immunosuppressive therapy during pregnancy, and an impaired functional renal reserve adaptation of the kidney allograft during pregnancy. Hyperfiltration (renal reserve capacity), defined as an eGFR increase during pregnancy of ≥15 mL/min or an eGFR increase of ≥20% from the pre-pregnancy eGFR, correlated significantly with stable allograft function after 24 months of follow-up. Moreover, in addition to the risk for an adverse pregnancy outcome, a negative eGFR slope 12 months before the start of pregnancy was also associated with a worsening allograft function at 24 months follow-up after the end of pregnancy.

The definition of preeclampsia in women with post-transplant pregnancy is challenging due to a pre-existing impaired allograft function with or without proteinuria and often underlying hypertension with the need of treatment. The incidence of pregnancies with adverse events was high in our cohort (40.9%), compared to 3–7% of regular pregnancies [21,22,23], but the numbers are comparable to other cohorts of women after KTX, with a rate of pregnancies with adverse events between 25% and 40% [3,7,24,25,26]. The change in new criteria for preeclampsia with severe features, also used in our study, adds a more precise definition of risk factors for adverse pregnancy outcomes. Beyond the established risk factors for adverse pregnancy outcomes, such as higher stages of chronic kidney disease (CKD), impaired allograft function at the time of pregnancy, uncontrolled hypertension, and pre-existing proteinuria [27,28], and instead of exclusively looking at non-dynamic kidney function parameters, we put our emphasis on dynamic changes in kidney function before and during pregnancy and up to 24 months after the end of pregnancy. Hyperfiltration during pregnancy develops early in gestation and persists until delivery [29,30]. Appropriate systemic vasodilation and the ability for renal hyperfiltration during pregnancy might play an important protective role during gestation [31,32,33,34], and previous or chronic damages to the allograft will likely impair physiologic pregnancy-related adaptions. Interestingly, Wiles et al. demonstrated in 178 women with CKD stages 3–5 that a gestational serum creatinine decrease of <10% compared to pre-pregnancy creatinine was a more significant risk predictor of early delivery and decline in kidney function than a higher CKD stage [35].

While many studies reported a worsening allograft function after pregnancy [19,36,37], case-control studies to compare kidney outcome results in pregnant women with non-pregnant women are rare [30,38]. Comparable allograft outcomes were reported in a few case-control studies with non-pregnant KTR [39,40,41], with patients with CKD of similar CKD stage [10], and with a non-transplanted population, mostly healthy, with two kidneys [6,30,42]. Here, we compared our findings in post-transplant pregnancies with a close 1:1 matched-pair group of kidney-transplanted women with no recent pregnancy to exclude confounders and bias. We observed a similar rate of change in eGFR in both post-transplant pregnancies and the control group until the end of pregnancy or observation and during follow-up for up to 24 months. As expected, proteinuria increased during pregnancy while it remained stable in the control group, but at 24 months follow-up, there were no more significant differences between the two groups. This is in line with other reports, where increased proteinuria resolved during long-term follow-up in most of the cases [43]. Gutierrez et al. reported proteinuria at the end of pregnancy in post-transplant women (0.8 ± 1.4 g/d) compared to a control group (0.1 ± 0.1 g/d) which persisted for up to one-year post-delivery but disappeared during the long-term follow up (0.3 ± 0.2 g/d). In contrast, higher proteinuria before or during pregnancy was associated with worse graft survival [44,45], but this could not be demonstrated in other studies [46].

To analyze risk factors in post-transplant patients with eGFR decrease over time, we did not see pregnancy itself as a risk contributor to worse allograft outcomes. Moreover, while parameters such as organ donor age, time since transplantation, immunosuppressive regimen, co-morbidities, and glomerulonephritis were not different between women with stable and worsening eGFR, we observed that the trajectory of eGFR before pregnancy was a risk predictor of worsening allograft function over time. Although only the eGFR slopes from 12 months before pregnancy until the start of pregnancy or, in the case of the controls, until the start of observation were significant, we assume that a continuous eGFR decline mirrors chronic allograft deterioration for various reasons (e.g., due to CNI toxicity or chronic allograft nephropathy) with the subsequent loss in kidney function in the long-term.

Little is known about the factors influencing an allograft´s ability to increase renal filtration during pregnancy and the impact of renal hyperfiltration on the pregnancy itself has been discussed controversially [10,47,48]. Park et al. reported a poor pregnancy outcome in 1931 women, of whom 94 had CKD, if eGFR levels during pregnancy remained below or raised above a reference level of 120–150 mL/min per 1.73 m^2^ [48]. Of these, women with a lower midterm eGFR (60–90 mL/min per 1.73 m^2^) experienced more frequent adverse pregnancy outcomes than did those with a higher midterm eGFR (90–120 mL/min per 1.73 m^2^). Remarkably, women after kidney transplantation with single kidneys do rarely reach supranormal eGFR levels, thus deleterious effects of very high eGFR increases are difficult to analyze in this special cohort. In contrast, Garg et al. demonstrated a higher rate of gestational hypertension or preeclampsia in women with previous living kidney donation than in matched non-donors with comparable health conditions (incidence, 11% vs. 5%, respectively) [46], which indicates a negative impact of a lower renal filtration reserve on subsequent changes in circulating plasma volume, placental perfusion, and endothelial damage. Bramham et al. did not see a correlation with adverse pregnancy outcomes despite the absence of a fall in creatinine in post-transplant pregnancies [6]. In contrast, Gosselink et al. showed, from a recent large nationwide cohort study with 177 patients, that a midterm percentage serum creatinine dip was significantly smaller in pregnancies with an adverse pregnancy outcome (mean difference of −4.5%) [49]. Our study also demonstrates that hyperfiltration (indicating the capacity to increase eGFR during pregnancy) occurred significantly more often and was more pronounced in the group with stable kidney function and in women without adverse pregnancy events. We assume that the ability for hyperfiltration reflects a reserved vascular and endothelial function of the kidney with the potential for hemodynamic adaptations during pregnancy. The relaxin pathway and the renin–aldosterone–angiotensin system, amongst others, likely contribute to the mechanism of blood flow increase and renal reserve capacity [32,50,51].

Kattah et al. reported results from allograft biopsies before and after pregnancy in kidney-transplanted women and observed more globally sclerosed glomeruli and an increase in chronic vascular injury [52]. Kolonko et al. suggested that hyperfiltration might damage allograft function by causing injury to the glomerulus with subsequent glomerulosclerosis [53]. The predictive value of a functional renal reserve in women with post-transplant pregnancies should be validated in larger prospective cohort studies. We suggest that women without adaptions of allograft eGFR during pregnancy need special attention regarding future complications such as worsening kidney function.

The immunosuppressive regimen in all KTR was dominated by the intake of CNI, MPA, and low-dose CS in both cohorts (the PG and CG). MPA was stopped in the case of a planned pregnancy, and patients were switched to azathioprin (in the case of triple IS) or to steroids in the case of previous dual IS with CNI and MPA. We hypothesized that the intake of CNI and the increased vascular stress of pregnancy might accelerate or aggravate allograft function or placental maturity, but we did not observe relevant differences, neither for the mother or fetal outcome nor for the long-term allograft outcome. Studies regarding the impact of IS on the complications of pregnancy in KTR are sparse. CNI are often the mainstay of immunosuppression in KTR. In general, the intake of CNI and azathioprin is considered to be relatively safe [30], but irrespective of pregnancy, CNI-induced decline in graft function in the long term is well described due to vasoconstriction associated with nephrotoxicity and hypertension [54,55]. Majak et al. reported that the use of cyclosporine in 175 pregnancies correlated significantly more frequently with pre-eclampsia [56]. Koenjer et al. retrospectively investigated the influence of CNI intake during pregnancy in 129 pregnancies with CNI and 125 pregnancies without. They did not find differences in major maternal or fetal outcomes between the two groups [57].

Remarkably, within the PG, we found that maintenance of a triple immunosuppressive regimen as a correlate for more intense immunosuppression posed a risk contributor for adverse pregnancy outcomes. Thus, the maintenance of dual immunosuppression during pregnancy might be beneficial, but this has to be validated using larger prospective studies.

Finally, we compared allograft losses after pregnancy with a matched-pair group of post-transplant women without pregnancy during the same observation period. Graft losses occurred late, several years after end of pregnancy or observation. We only experienced one pregnancy-related graft loss shortly after delivery due to severe chronic allograft nephropathy and fibrosis of the allograft. This patient had, already at the start of pregnancy, a severely diminished allograft function without any renal reserve capacity and a continuous decline in eGFR from 24 months before pregnancy onwards. The majority of data derived from kidney transplant recipient cohorts also showed a good outcome and no increased risk of rejection, allograft loss after pregnancy, or impairment in kidney function [6,58]. Another concern is the development of de novo DSA after pregnancy. We did not observe an increased frequency of de novo DSA after delivery in our cohort, which needs to be confirmed using larger studies. Levidiotis et al. demonstrated long-term follow-up data over 40 years of transplant patients with pregnancies that did not worsen fetal and maternal outcomes when compared to transplant patients without pregnancies [59]. This is also in line with several studies that have evaluated the impact of pregnancy on long-term graft function and survival [30].

The limitation of our study is its retrospective nature with a small single-center cohort of pregnancies. Except for the deterioration in eGFR before pregnancy, we did not find any other predictors for adverse pregnancy outcomes. The strength of this study is the comparison with a matched control cohort with detailed information regarding baseline characteristics, allograft function parameters with the emphasis on dynamic changes, and a detailed long-term mean follow-up time in both cohorts including regular determinations of HLA antibodies [60]. Since common clinical indicators were not helpful for risk prediction, the use of current biomarkers such as sFlt-1 and PlGF and the detection and validation of new biomarkers derived from multiple organ and cellular sources need to be explored in future studies [61].

## 5. Conclusions

Pregnancies in women after KTX showed good maternal and allograft outcomes. Pre-pregnancy factors, except for a declining renal allograft function in the year before pregnancy, were not helpful to predict the outcomes. The ability of the allograft to perform hyperfiltration during pregnancy as an indicator of preserved renal reserve capacity correlated with a more favorable pregnancy outcome and stable kidney function in the long term. In our small cohort, there was no obvious pregnancy-related alloimmunization against the allograft.

## Figures and Tables

**Figure 1 jcm-12-01545-f001:**
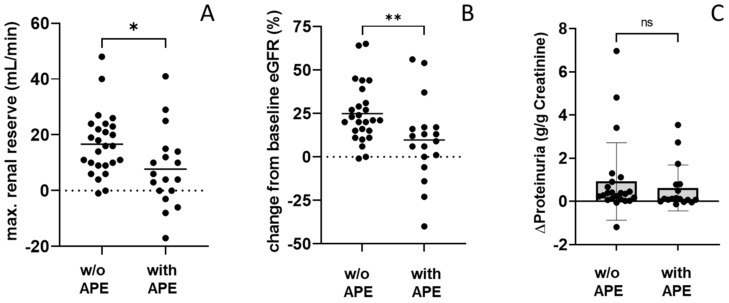
Impact of hyperfiltration during pregnancy and change from baseline eGFR on adverse pregnancy outcome. Maximal (max.) renal reserve expressed as mL/min (**A**) or as the percentage of increase from baseline eGFR (**B**) correlated with a better pregnancy outcome. Change in proteinuria (g/g creatinine) did not differ between pregnancies with and without (w/o) adverse pregnancy events (APE) (**C**). * = *p* < 0.05, ** = *p* < 0.01, ns = not significant.

**Figure 2 jcm-12-01545-f002:**
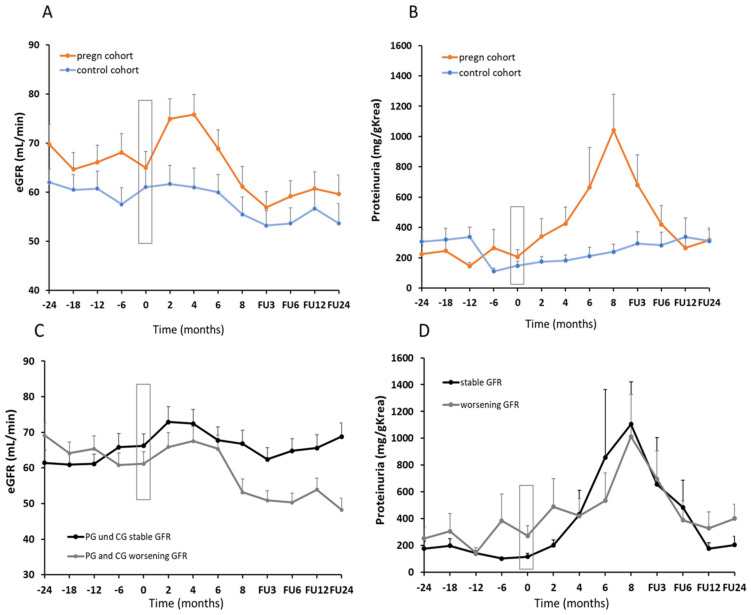
GFR slopes pre-pregnancy and follow-up in women with post-transplant pregnancies (pregn cohort) compared to a control cohort. GFR slopes (**A**) and the course of proteinuria (**B**) in all post-transplant pregnancies and matched controls. GFR slopes in all kidney transplant recipients (KTR) subdivided into KTR women with stable (black line) or worsening (gray line) allograft function over time (**C**). Change in proteinuria during post-transplant pregnancies in women with stable (black line) or worsening (gray line) allograft function over time (**D**).

**Figure 3 jcm-12-01545-f003:**
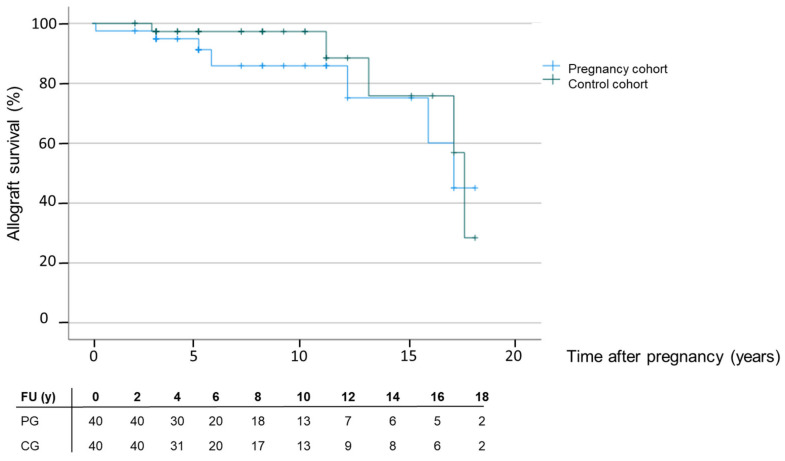
Kaplan–Meier estimates of allograft survival in women with post-transplant pregnancies (blue line) versus women without post-transplant pregnancies (green line) during the same observation period.

**Table 1 jcm-12-01545-t001:** Baseline characteristics of KTR with pregnancy matched to a control cohort.

Patient Characteristics *	Pregnancy Group *n* = 40	Control Group *n* = 40	Statistical Group Difference (*p*-Value)
**Age at pregnancy (y)**	32.5 ± 4.4	32.9 ± 4.9	n.s.
**Age at TX (y)**	25.6 ± 6.9	27.2 ± 6.4	n.s.
**Kidney Disease**			n.s.
IgA nephropathy	11	5	
Other glomerulonephritis	6	8	
Diabetic nephropathy	4	2	
Polycystic kidney disease	0	2	
Reflux or hypoplastic kidney disease	4	7	
Other reasons	6	10	
Unknown	9	5	
**Type of transplant**			n.s.
Single kidney	37	38	
Combined pancreas–kidney	3	2	
**Time between kidney failure and transplantation (m)**	34.8 ± 48.2	37.8 ± 42.3	n.s.
**Time after transplantation (y)**	5.8 ± 4.8	4.6 ± 4.1	n.s.
**Donor age (y)**	39.6 ± 17.2	47.6 ± 9.3	0.005
**Type of donation**			n.s.
Postmortem	20	18	
ABO compatible living donation	18	20	
ABO incompatible living donation	2	2	
**Immunosuppressive regimen**			
Triple IS	18	26	0.072
Dual IS	21	13	0.07
Steroids	26	28	0.633
Calcineurin inhibitor	35	39	0.201
Mycophenolic acid	29	33	0.284
Others (mTORi, belatacept, or azathioprine)	7	4	0.33
**Induction therapy at TX**			n.s.
Basiliximab	25	20	
Thymoglobulin	4	2	
HLA class I at time of pregnancy	8	10	n.s.
HLA class II at time of pregnancy	7	7	
DSA	2/35	4/40	
**Co-morbidities**			n.s.
Hypertension	24	22	
Diabetes mellitus	4	7	
Systolic BP (mmHg)	123 ± 12	120 ± 11	n.s.
Diastolic BP (mmHg)	84 ± 9	81 ± 8	
**History of pregnancy**		N/A	
Previous successful pregnancy	10		
Previous miscarriage	15		
**Kidney function at start of pregnancy/observation**			n.s.
eGFR (mL/min)	64 ± 22	61 ± 22	
Creatinine (mg/dL)	1.2 ± 0.5	1.3 ± 0.4	
Proteinuria (mg/g creatinine)	218 ± 278	186 ± 163	

* Total numbers as “*n*” unless otherwise specified. KTR = kidney transplant recipients; TX = transplantation; IS = immunosuppression; HLA = human leukocyte antigen; DSA = donor-specific antibodies; BP = blood pressure; mTORi = mammalian target of rapamycin inhibitor; eGFR = estimated glomerular filtration rate; m = months; y = years.

**Table 2 jcm-12-01545-t002:** Pregnancy and allograft outcomes in post-transplant pregnancies.

Patient Characteristics *	Pregnancy Group *n* = 40
**Pregnancy ≥ 12 weeks**	38
**Median gestational weeks (IQR)**	35.15 (31.5, 37.225)
**Adverse pregnancy outcome**	18
Severe hypertension **	3
Acute kidney injury (AKIN) II	6
Acute kidney injury (AKIN) III	2
Gestational thrombocytopenia	12
Abortion ≥ 12 weeks	4
Stillbirth	1
Early preterm delivery ≤ 32 weeks	3
Intrauterine growth restriction	8
**Kidney function at end of pregnancy**	
eGFR (mL/min)	57.5 ± 27.1
Proteinuria (mg/g creatinine)	1051 ± 1541
>500 mg/g creatinine (%)	42.5
>1000 mg/g creatinine (%)	30
ΔeGFR (mL/min)	−5.5 ± 17.7
ΔProteinuria (mg/g creatinine)	830 ± 1498
Deterioration of eGFR > 5 mL/min (%)	50

* Total numbers as “*n*” unless otherwise specified. ** Severe hypertension defined as systolic RR > 160 mmHg or diastolic RR > 110 mmHg. Gestational thrombocytopenia defined as a platelet count below 150 × 109/L. Intrauterine growth restriction defined as growth <10th percentile after gestational week 24 + 0. eGFR = estimated glomerular filtration rate.

**Table 3 jcm-12-01545-t003:** Characteristics stratified by status for adverse pregnancy events.

Variable *	Pregnancy Group Wo APE *n* = 26	Pregnancy Group with APE *n* = 18	Statistical Group Difference (*p*-Value)
**Age at pregnancy (y)**	32.7 ± 4.3	33.1 ± 4.3	n.s.
**Age at TX (y)**	25.8 ± 7.2	25.3 ± 6.7	n.s.
**Time after TX (y)**	5.6 ± 4.4	6.8 ± 5.3	n.s.
**Donor age (y)**	39.3 ± 17.4	39.7 ± 19.4	n.s.
**Living donation**	15	6	n.s.
**Systolic BP (mmHg)**	124 ± 11	127 ± 14	n.s.
**Diastolic BP (mmHg)**	84 ± 10	86 ± 13	
**Previous pregnancy**	12	8	n.s.
**Previous miscarriage**	12	7	n.s.
**Basal IS**			
Triple IS	9	9	n.s.
Dual IS	15	9	
Steroids	14	13	
Calcineurin inhibitor	23	16	
Mycophenolic acid	16	15	
mTOR inhibitor	4	2	
Azathioprin	1	0	
**IS during pregnancy**			
Triple IS	2	7	**0.021**
Dual IS	23	10	**0.031**
Steroids	23	17	n.s.
Calcineurin inhibitor	25	17	n.s.
Mycophenolic acid	0	1	n.s.
Azathioprin	5	6	n.s.
**Co-morbidities**			
Chronic hypertension	14	8	n.s.
Diabetes mellitus	1	4	
Diabetes during pregnancy	2	5	
**Weeks of gestation**	36.7 ± 2.2	29.4 ± 7.3	**<0.001**
**Childbirth parameters**			
Newborn weight	2514 ± 524	1566 ± 762	**<0.001**
Newborn height	47.3 ± 3.3	39.5 ± 7.1	**<0.001**
**Allograft function at pregnancy**			
**eGFR** (mL/min)			
at start	64.5 ±16.3	61.5 ±23.6	n.s.
month 3	77.9 ± 22	69.1 ± 30.3	n.s.
at end of pregnancy	61.7 ± 24.9	53.28 ± 31.29	n.s.
pre-pregn eGFR < 60	11	7	n.s.
Hyperfiltration during pregn	15	5	n.s.
Renal reserve capacity (mL/min)	16.6 ± 11.3	7.8 ± 13.9	**0.021**
eGFR increase (%)	25 ± 17	10 ± 24	**0.008**
Increase >20% (%)	65%	17%	**0.002**
**Proteinuria** (mg/g creatinine)			
at start	240 ± 306	172 ± 222	n.s.
at end of pregnancy	1176 ± 1775	780 ± 1109	n.s.
Rise during pregn >500 mg/g creatinine	8	7	n.s.
**Fetal ultrasound**			
PI uterine artery			
week 22–23	1.02 ± 0.27	1.10 ± 0.31	n.s.
week 24–26	0.81 ± 0.20	1.10 ± 0.37	**0.027**
week 27–30	0.83 ± 0.40	0.93 ± 0.35	n.s.
PI umbilical artery			
week 22–23	1.13 ± 0.18	1.25 ± 0.39	n.s.
week 24–26	1.05 ± 0.26	1.60 ± 1.24	n.s.
week 27–30	1.05 ± 0.17	1.19 ± 0.54	n.s.
Corticosteroid pulse peripartum	7	2	n.s.
**sFlt-1/PlGF** (highest value)			
at 2nd trimester (*n*)	8.4 ± 9.5 (8)	54.3 ± 55.2 (4)	0.085
at 3rd trimester (*n*)	65.4 ± 76.3 (7)	68.4 ± 43.6 (7)	0.528

* Total numbers as “*n*” unless otherwise specified. APE = adverse pregnancy events; TX = transplantation; IS = immunosuppression; BP = blood pressure, mTOR = mammalian target of rapamycin; eGFR = estimated glomerular filtration rate; pregn = pregnancy; PI = pulsatility index; sFlt/PlGF = soluble fms-like tyrosine kinase-1/placental growth factor ratio; y = years.

**Table 4 jcm-12-01545-t004:** Allograft outcome in post-transplant pregnancies and controls.

Patient Characteristics	Pregnancy Group *n* = 40	Control Group *n* = 40	Statistical Group Difference (*p*-Value)
**12 months FU**			
eGFR (mL/min)	59.1 ± 21.5	56.6 ± 23.2	n.s.
Proteinuria (mg/g creatinine)	268 ± 474	337 ± 741	n.s.
>500 mg/g creatinine (%)	9.5	14.3	n.s.
>1000 mg/g creatinine (%)	4.8	3.2	n.s.
ΔeGFR(mL/min)	−4.2 ± 9.6	−4.5 ± 13.4	n.s.
ΔProteinuria (mg/g creatinine)	52 ± 487	172 ± 694	n.s.
Deterioration in eGFR > 5 mL/min (%)	54.5	32.5	n.s.
**24 months FU**			
eGFR (mL/min)	57.7 ± 24.5	53.6 ± 23.4	n.s.
Proteinuria (mg/g creatinine)	321 ± 442	308 ± 506	n.s.
>500 mg/g creatinine (%)	17.1	16.2	n.s.
>1000 mg/g creatinine (%)	9.8	10.8	n.s.
ΔeGFR	−5.4 ± 14.3	−7.6 ± 14.1	n.s.
ΔProteinuria	143 ± 424	169 ± 446	n.s.
Deterioration in eGFR > 5 mL/min (%)	56.8	57.8	n.s.
Renal graft loss	1	0	n.s.

eGFR = estimated glomerular filtration rate; FU = follow up (time point after the end of pregnancy/observation).

**Table 5 jcm-12-01545-t005:** Characteristics stratified by status for adverse pregnancy events.

Variable *	Stable eGFR *n* = 36	Worsening eGFR *n* = 49	Statistical Group Difference (*p*-Value)
**Pregnancy**	20	26	n.s.
**Pregnancy with APE**	6/20	12/26	
**Age at TX**	26.8 ± 5.8	25.9 ± 7.4	n.s.
**Living donation**	20	25	n.s.
**Donor age (y)**	43 ± 16	43 ± 4	n.s.
**Time between kidney failure and TX (m)**	43.8 ± 54.6	37.2 ± 43.3	n.s.
**Time after TX (y)**	4.7 ± 4.2	5.7 ± 4.7	n.s.
**GN as kidney disease**	14	15	n.s.
**Basal IS**			
Steroids	26	30	n.s.
Calcineurin inhibitor	34	44	
Mycophenolic acid	27	38	
**Co-Morbidities**			
Hypertension	20	22	n.s.
Diabetes mellitus	3	8	
**Kidney function**			
Baseline eGFR	66.3 ± 19.2	61.2 ± 23.3	n.s.
ΔeGFR −12 months to start	4.9 ± 12.0	−3.9 ± 11.0	**0.001**
ΔeGFR −18 months to start	4.6 ± 18.8	−1.4 ± 18.1	**0.127**
ΔeGFR −24 months to start	−0.6 ± 20.6	−7.5 ± 14.7	n.s.
Baseline proteinuria (mg/g creatinine)	152 ± 19	213 ± 254	n.s.
Proteinuria at end of pregnancy/observation (mg/g creatinine)	277 ± 460	340 ± 482	n.s.
**DSA pre**	4	3	n.s.
**DSA 24FU**	6	3	
**eGFR variability during pregnancy**			
Hyperfiltration during	15/20	6/26	**0.001**
pregn (of *n* pregn)			
Renal reserve capacity	18.4 ± 13.1	8.7 ± 11.7	**0.002**
(mL/min)			
Increase from baseline	27 ± 23	12 ± 17	**0.002**
eGFR (%)			
Increase >20% (*n*)	15/20	6/26	**0.001**
**Follow up of allograft function**			
ΔeGFR 12 m FU	0.1 ± 8.3	−7.6 ± 12.8	**0.004**
ΔeGFR 24 m FU	3.3 ± 11.2	−13.3 ± 11.4	**<0.01**
Stable proteinuria 12 m FU	31/34	42/48	n.s.
Stable proteinuria 24 m FU	28/34	39/47	n.s.

* Total numbers as “*n*” unless otherwise specified. KTR = kidney transplant recipients; APEs = adverse pregnancy events, eGFR = estimated glomerular filtration rate; stable proteinuria defined as change in proteinuria < 300 mg/g creatinine from baseline; y = years; m = months; GN = glomerulonephritis; IS = immunosuppression; TX = transplantation; DSA = donor specific antibodies; preng = pregnancy; FU = follow-up. Hyperfiltration defined as eGFR increase ≥ 15 mL/min during pregnancy.

## Data Availability

The data presented in this study are available on request from the corresponding author. The data are not publicly available.

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
