# Peer review of "Pregnancy after Kidney Transplantation—Impact of Functional Renal Reserve, Slope of eGFR before Pregnancy, and Intensity of Immunosuppression on Kidney Function and Maternal Health"

_jcm, 2023, doi:10.3390/jcm12041545_

Round 1

Reviewer 1 Report

Very informative and well written- I really enjoyed reading.  I believe both providers and their patients will benefit greatly from this research.  

Consider adding more to the discussion that hypothesizes why you found certain things and less of re-stating your results- eg why would the neg 12m slope pre-pregnancy matter but not the 18m or 24m? were they switching away from MMF? why does hyper-filtration matter? is it a marker for vascular reserve/ endothelial function? if common clinical indicators aren't helpful for risk prediction, should we be looking more into biomarkers?

For Table 2: I would consider breaking this into two separate tables.  One table could be descriptive of pregnancy outcomes of pregnancy group alone- and include here their kidney outcomes at the end of pregnancy as well.  I don't think it's appropriate to compare kidney outcomes at end of pregnancy to those who did not experience pregnancy.  The second table could then include the comparisons of 12m and 24m follow-up. 

Cr/eGFR at start of pregnancy/observation - what was the mean or median gestational age for this?  There is an increase in eGFR fairly early on in pregnancy so it would be important to know and could potentially affect how the renal reserve capacity was calculated. Would it make more sense to use Cr/eGFR perhaps ~3m prior to pregnancy?

For table 3: include N's for the sFlt-1/PlGF data or consider just including it in the text since numbers were so low. 

Two additional references that would be of value to include:

Wiles K, Webster P, Seed PT, Bennett-Richards K, Bramham K, Brunskill N, Carr S, Hall M, Khan R, Nelson-Piercy C, Webster LM, Chappell LC, Lightstone L. The impact of chronic kidney disease Stages 3-5 on pregnancy outcomes. Nephrol Dial Transplant. 2021 Nov 9;36(11):2008-2017. doi: 10.1093/ndt/gfaa247.

Harel Z, McArthur E, Hladunewich M, Dirk JS, Wald R, Garg AX, Ray JG. Serum Creatinine Levels Before, During, and After Pregnancy. JAMA. 2019 Jan 15;321(2):205-207. doi: 10.1001/jama.2018.17948.

Author Response

To reviewer 1:

We are truly thankful for the reviewer´s positive and helpful comments.

Here is our point-by-point response.

Reviewer 1: "Consider adding more to the discussion that hypothesizes why you found certain things and less of re-stating your results- eg why would the neg 12m slope pre-pregnancy matter but not the 18m or 24m? were they switching away from MMF? why does hyper-filtration matter? is it a marker for vascular reserve/ endothelial function? if common clinical indicators aren't helpful for risk prediction, should we be looking more into biomarkers?"

Response: We appreciate the reviewer´s comment. We agree and added several paragraphs within the discussion to interpret our results.

We now changed this into:

  • Moreover, while parameters such as organ donor age, time since transplantation, immunosuppressive regimen, co-morbidities, and glomerulonephritis were not different between women with stable and worsening eGFR, we observed, that the trajectory of eGFR before pregnancy was a risk predictor of worsening allograft function over time. Although, only eGFR slopes from 12 months before pregnancy until start of pregnancy or, in case controls, until start of observation were significant, we assume, that a continuous eGFR decline mirrors chronic allograft deterioration for various reasons (e.g. due to CNI toxicity or chronic allograft nephropathy) with subsequent loss of kidney function in the long-term.
  • Our study also demonstrates that hyperfiltration (indicating the capacity to increase eGFR during pregnancy) occured significantly more often and was more pronounced in the group with stable kidney function and in women without adverse pregnancy events. We assume, that the ability for hyperfiltration reflects a reserved vascular and endothelial function of the kidney with the potential of hemodynamic adaptations during pregnancy. The relaxin pathway and the renin-aldosterone-angiotensin system, amongst others, likely contribute to the mechanism of blood flow increase and renal reserve capacity (Conrad KP: J Soc Gynecol Investig 11: 438–448, 2004; Cadnapaphorn-chai MA, et al., Am J Physiol Renal Physiol 280: F592–F598, 200116; Davison JM, et al., J Am Soc Nephrol 15: 2440–2448, 2004).
  • Since common clinical indicators were not helpful for risk prediction, the use of current biomarkers such as sFlt-1 and PlGF and the detection and validation of new biomarkers derived from multiple organ and cellular sources need to be explored in future studies (MacDonald et al., eBioMedicine, Volume 75, January 2022, 103780).

Reviewer: "For Table 2: I would consider breaking this into two separate tables.  One table could be descriptive of pregnancy outcomes of pregnancy group alone- and include here their kidney outcomes at the end of pregnancy as well.  I don't think it's appropriate to compare kidney outcomes at end of pregnancy to those who did not experience pregnancy.  The second table could then include the comparisons of 12m and 24m follow-up. "

Response: we thank the reviewer for his/her helpful comment. We now separated Table 2 into two separate tables as suggested.

Reviewer: "Cr/eGFR at start of pregnancy/observation - what was the mean or median gestational age for this?  There is an increase in eGFR fairly early on in pregnancy so it would be important to know and could potentially affect how the renal reserve capacity was calculated. Would it make more sense to use Cr/eGFR perhaps ~3m prior to pregnancy?"

Reponse: We thank for the reviewer´s comment.

Gestational age was calculated starting from the first day of the last menstrual period, thus in principle approximately 2 weeks before conception. We calculated the increase of eGFR from a mean pre-pregnancy eGFR, which was determined by the mean of at least two eGFR measures prior to conception (around -6, -12 month before conception and around the first day of the last menstrual period). We now added this information to the method section.

For table 3: include N's for the sFlt-1/PlGF data or consider just including it in the text since numbers were so low. 

Reponse: We thank for the reviewer´s comment.  We now added N´s within Table 3.

Reviewer: "Two additional references that would be of value to include:

Wiles K, Webster P, Seed PT, Bennett-Richards K, Bramham K, Brunskill N, Carr S, Hall M, Khan R, Nelson-Piercy C, Webster LM, Chappell LC, Lightstone L. The impact of chronic kidney disease Stages 3-5 on pregnancy outcomes. Nephrol Dial Transplant. 2021 Nov 9;36(11):2008-2017. doi: 10.1093/ndt/gfaa247.

Harel Z, McArthur E, Hladunewich M, Dirk JS, Wald R, Garg AX, Ray JG. Serum Creatinine Levels Before, During, and After Pregnancy. JAMA. 2019 Jan 15;321(2):205-207. doi: 10.1001/jama.2018.17948."

Reponse: We agree and added both references to the manuscript.

Reviewer 2 Report

This manuscript provides very useful data regarding the pregnancy outcomes of women with a kidney transplant. Methods and criteria of judgment are not only appropriate but also modern. It is well-written, and straightforward. References are up to date.

I have no modifications to suggest, but only a question : did the authors collect the information about spontaneous vs. ART-induced pregnancies ? this might have some importance since one of the main results from their study is that the trajectory of creatinine throughout pregnancy (renal reserve, mostly due to the production of relaxin) is associated with adverse pregnancy outcomes.

Author Response

To Reviewer 2

We are truly thankful for the reviewer´s positive comments.

Reviewer: "This manuscript provides very useful data regarding the pregnancy outcomes of women with a kidney transplant. Methods and criteria of judgment are not only appropriate but also modern. It is well-written, and straightforward. References are up to date.

I have no modifications to suggest, but only a question : did the authors collect the information about spontaneous vs. ART-induced pregnancies ? this might have some importance since one of the main results from their study is that the trajectory of creatinine throughout pregnancy (renal reserve, mostly due to the production of relaxin) is associated with adverse pregnancy outcomes."

Reponse: We thank the reviewer for her/his comments. We agree, that ART-induced pregnancies might have a negative impact on pregnancy outcome and kidney function. Unfortunately, the documentation of ART-induced pregnancies in our study is insufficient for analysis.